# An Overview on the Adhesion Mechanisms of Typical Aquatic Organisms and the Applications of Biomimetic Adhesives in Aquatic Environments

**DOI:** 10.3390/ijms25147994

**Published:** 2024-07-22

**Authors:** Jiani Liu, Junyi Song, Ling Zeng, Biru Hu

**Affiliations:** College of Science, National University of Defense Technology, Changsha 410073, China

**Keywords:** aquatic environments, biological models, adhesion mechanism, bioadhesives, biomimetic adhesives

## Abstract

Water molecules pose a significant obstacle to conventional adhesive materials. Nevertheless, some marine organisms can secrete bioadhesives with remarkable adhesion properties. For instance, mussels resist sea waves using byssal threads, sandcastle worms secrete sandcastle glue to construct shelters, and barnacles adhere to various surfaces using their barnacle cement. This work initially elucidates the process of underwater adhesion and the microstructure of bioadhesives in these three exemplary marine organisms. The formation of bioadhesive microstructures is intimately related to the aquatic environment. Subsequently, the adhesion mechanisms employed by mussel byssal threads, sandcastle glue, and barnacle cement are demonstrated at the molecular level. The comprehension of adhesion mechanisms has promoted various biomimetic adhesive systems: DOPA-based biomimetic adhesives inspired by the chemical composition of mussel byssal proteins; polyelectrolyte hydrogels enlightened by sandcastle glue and phase transitions; and novel biomimetic adhesives derived from the multiple interactions and nanofiber-like structures within barnacle cement. Underwater biomimetic adhesion continues to encounter multifaceted challenges despite notable advancements. Hence, this work examines the current challenges confronting underwater biomimetic adhesion in the last part, which provides novel perspectives and directions for future research.

## 1. Introduction

Adhesion typically involves cohesion within the adhesive network and bond formation at the interface between the adhesive and the attachment surface. The recent advancements in tissue engineering and implantable healthcare monitors have led to a significant focus on adhesives that can adapt to aquatic environments. However, the adhesion is hindered more in aquatic environments than in dry conditions due to water or other liquids [1,2,3,4,5]. The influence of water on adhesives is primarily evident in four aspects: (1) Water molecules adsorbed on the adherend interface obstruct the molecular contacts between the adhesive and the substrate, which impedes the development of interfacial adhesion [6,7,8]. (2) The infiltration of water molecules to the adherend interface results in the formation of a water film between the adhesive and the substrate. Consequently, interfacial adhesion is disrupted [9,10]. (3) The hydrolytic degradation of adhesives in water or other aquatic environments undermines their cohesive structure and stability [11,12]. (4) The water absorption of adhesives reduces their mechanical strength, which increases the probability of cohesive failure [5].

Engineering adhesives utilized in aquatic environments primarily include epoxy resins, urea-formaldehyde resins, phenolic resins, and polyurethanes [13,14,15,16,17]. However, their applications in some special fields (e.g., biomedical) encounter challenges such as prolonged curing time, subpar adhesion performance, restricted reusability, and the potential for leaching toxicity. Additionally, FDA-certified biomedical tissue adhesives exhibit their limitations. For example, cyanoacrylate tissue adhesives (e.g., Dermabond) display heightened brittleness after curing, which mismatches the toughness of tissue interfaces. In addition, their degradation byproducts may induce inflammatory responses [18]. The degradation rate of fibrin tissue adhesives is excessively rapid and their adhesion strength is low (<0.2 MPa). Polyethylene glycol (PEG) adhesives (e.g., CoSeal and DuraSeal) manifest a significant swelling rate, reaching approximately 400% [19,20]. Therefore, it is of significance to develop underwater adhesives featuring fast curing, superior adhesion, and minimal toxicity.

Natural adhesives secreted by marine organisms (e.g., mussels, sandcastle worms, and barnacles) exhibit efficient adhesion in aquatic environments. For instance, the plaque-thread system of a single mussel byssal thread demonstrates an adhesive strength of approximately 0.5 N, with an average bonding strength reaching up to 200 kPa [21]. Barnacles, with a diameter of 8 cm, exhibit an adhesion strength of about 0.4 MPa [22]. Barnacle cement remains firmly adhered to the substrate surface even in the event of barnacle mortality or forced detachment of their shells. Additionally, biological adhesives typically demonstrate a greater resistance to swelling in aquatic environments compared to synthetic adhesives. Moreover, biological materials show a relatively superior biocompatibility due to the lack of specific chemical components [23]. Consequently, exploring the adhesion mechanisms of marine organisms can facilitate the development of novel adhesion technology in aquatic environments [24].

The underwater adhesion processes of mussels, sandcastle worms, and barnacles were initially introduced, with a focus on the structure and components of their secreted biological adhesives. Subsequently, the adhesion mechanisms of these biological adhesives were summarized by analyzing the physicochemical processes of adhesion formation in these three marine organisms. A vast array of biological adhesive designs inspired by these marine organisms was then presented, with an emphasis on the correlation between adhesive effects and aquatic environments. Finally, the potential applications and existing challenges in engineering and biomedical fields were comprehensively scrutinized based on the mechanisms of biological adhesives.

## 2. Biological Models and Underwater Adhesion Processes

Mussels, sandcastle worms, and barnacles can form adhesions to various surfaces by secreting bioadhesives. These bioadhesives maintain effective adhesion with minimal volume changes despite the erosion of seawater [25,26]. The phenomenon is closely related to the secretion, solidification, and structure of bioadhesives.

### 2.1. Mussels

Mussels inhabit the intertidal zone. They utilize more than fifty byssal threads to anchor themselves to rocks and withstand the scouring force of tides. The ventral groove extends from the foot belly to form a cavity and tightly adheres to the substrate upon finding a suitable attachment point using the mussel foot. Then, glands surrounding the ventral groove, including the collagen gland, accessory gland, and phenol gland, secrete various substances (e.g., mussel adhesive proteins (MAPs)), which are injected into the cavity. The secretions undergo a series of physicochemical changes and transform into byssal threads as the physiological environment within the cavity rapidly transitions to seawater. The whole process can be completed within minutes [27,28,29].

Mussel byssal threads primarily rely on the plaque-thread system for robust underwater adhesion from the structural perspective. The plaque-thread system can be divided into three components, that is, the core, cuticle, and plaque. Each possesses distinct compositions, microstructures, and functions (Figure 1A–C) [30]. The core mainly consists of semi-crystalline collagenous proteins (preCols), which form high-performance byssal thread fibers via self-assembly [30,31]. The linear arrangement of fibers along the axial direction can be observed by scanning electron microscopy (SEM) [32]. The non-collagenous preCol domains within the core structure function as the primary load-bearing units of the byssal thread. PreCols provide the core with an initial stiffness of approximately 900 MPa, in addition to significant toughness, ductility, and a certain degree of self-repairing capability under cyclic loading [33,34,35,36]. The primary function of the cuticle is to protect the core. The cuticle is approximately 2–5 μm-thick, with around 50% of its surface layer being covered by particles measuring about 0.8 μm in diameter. Therefore, it is extremely hard and ductile [30,37,38]. The plaque at the base of the byssal thread is primarily responsible for interfacial adhesion. It features a foam-like porous microstructure with pore diameters predominantly measuring below 2 μm [39]. The porous structure influences crack blunting at the crack tip, which hinders crack propagation [40]. The adhesion strength of each plaque is approximately 5.7–6 MPa [41]. The formation of the plaque primarily involves five steps: (1) A negative pressure is created near the substrate to form a cavity. (2) The chemical environment is regulated within the cavity through pH, ion strength, and redox potential. (3) Soluble proteins (coagulants) are secreted in the cavity. (4) Water/protein phase reversal occurs. (5) The solidification is enhanced by crosslinking reactions facilitated by minerals and enzymes [7,8,32].

In conclusion, the mussel’s plaque-thread system, consisting of the core, cuticle, and plaque, facilitates byssal threads to possess exceptional mechanical strength, robust cohesion, toughness, and ductility (reaching up to 327 ± 32% of their original length). On the other hand, the interactions between the plaque and the substrate promote robust adhesion of byssal threads to virtually any surface in wet conditions [42,43].

### 2.2. Sandcastle Worms

Sandcastle worms can thrive in the intertidal zone by secreting sandcastle glue. The glue rapidly binds sand grains and biogenic minerals in the marine environment to construct a sturdy tubular shell as a shelter. The adhesion process mainly consists of the following stages: (1) Sandcastle worms use ciliated tentacles to capture sand grains and other small particles from turbulent water columns. Then, these grains and particles are deployed around their bodies within the building organ equipped with numerous cilia. The cilia are responsible for detecting and locating particles of suitable size, shape, and surface chemistry (Figure 1D) [44]. (2) The surface of the building organ features numerous paired micropores, which are internally connected to glands that secrete sandcastle glue. The glands contain two types of secretory cells that release homogeneous or heterogeneous secretory granules, respectively. The secretory granules are then transported to the paired micropores via axon-like cell structures (Figure 2B) [45]. (3) Fine gaps among the particles facilitate the uptake of two different types of secretory granules through capillary action. Concave meniscus bridges are formed due to the lower Laplace pressure of sandcastle glue compared to the surrounding water phases (Figure 1E,F). (4) The two different types of secretory granules rapidly rupture within a short period (approximately a few seconds) in response to environmental changes or mechanical stimulation. Secretory granules fuse with surrounding granules and simultaneously solidify. However, the individual integrity of each secretory granule is largely maintained. The interactions among the multiple components of sandcastle glue ultimately form a porous structure resembling foam. These stages collectively contribute to the formation of sandcastle glue and the subsequent construction of tubular shells for sandcastle worms (Figure 1G) [7,46,47].

### 2.3. Barnacles

Barnacles can firmly attach their calcium baseplates to various substrates (e.g., natural rocks, artificial surfaces, and animal skins) by secreting barnacle cement (Figure 1H). The lifecycle of a barnacle includes four stages: nauplius, cyprid, juvenile, and adult. Cyprids release temporary reversible adhesives during surface exploration. They produce permanent adhesives (i.e., cyprid cement) after the section of a suitable location [48]. Barnacles secrete barnacle cement to maintain firm attachment when they develop into adults [49]. The work specifically examined the underwater adhesion of acorn barnacle cement as a representative example.

Adult barnacles initially extend their lateral plates to establish an interspace between the barnacle baseplate and the attachment surface when they initiate adhesion to a substrate. Subsequently, they secrete a lipid-rich matrix or an oxidizing liquid to displace the interfacial water film and other contaminants, which cleans the substrate. Afterward, barnacle cement is secreted for strong adhesion [50,51,52]. The synthesis and storage of barnacle cement occur in glands located near the baseplate. Subsequently, the barnacle cement is transported to the vicinity of the substrate through extracellular drainage channels [53]. The substance is then transferred to the matrix through sliding movements of the cuticle [52]. Barnacle cement may undergo mild crosslinking or possess adhesion properties during storage and transportation according to some research [49,54].

Microscopic analysis of barnacle cement reveals the presence of numerous fibrous structures (Figure 1I–K), including micrometer-scale rods, globular structures, and a matrix composed of fibrils. The majority of solidified barnacle cement consists of fibrous structures with diameters ranging from several nanometers to tens of nanometers. The analysis of barnacle cement using far-UV CD, FTIR, and ThT tests reveals the presence of amyloid-like protein domains [26,55,56,57]. The amyloid-like proteins possess robust stability and exceptional strength, which induce the conversion of soluble proteins into insoluble states [12,58]. Fibrous structures are crucial for the solidification of barnacle cement, as well as for imparting robust adhesion, mechanical properties, and stability based on an increasing number of studies.

**Figure 1 ijms-25-07994-f001:**
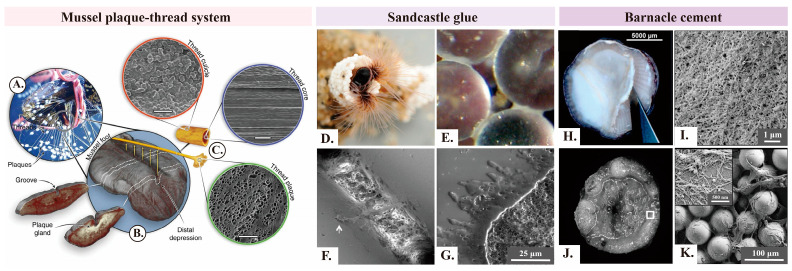
Microscopic images of mussel byssal threads, sandcastle glue, and barnacle cement (**A**–**C**). Macroscopic and microscopic structures of mussel byssal threads. (**A**) Mussel byssal threads adhere to the surface of organic glass. (**B**) μ-CT images of iodine-stained mussel byssal threads on the ventral surface show the internal gland and groove structures near the middle and distal depressions of the foot, respectively. (**C**) Schematic diagram of a single byssal thread, with scanning electron microscope images showing the morphology of the cuticle (scale bar: 3 μm), core (scale bar: 5 μm), and plaque (scale bar: 50 μm) [30]. Copyright 2004, Wiley-VCH. (**D**–**G**) Morphology of sandcastle glue. (**D**) Laboratory conditions: Sandcastle worms construct shelters using zirconia beads. (**E**,**F**) Fine gaps among particles inhale sandcastle glue to form concave capillary bridges. (**G**) Scanning electron microscope images of fractured glue after freeze-drying [46]. Copyright 2013, American Chemical Society. (**H**–**K**) Nanofiber structure of barnacle cement. (**H**) Thick and opaque barnacle cement. (**I**) Scanning electron microscopy reveals abundant small nanofibers within (**H**). (**J**) Barnacles are placed on a substrate containing numerous glass beads for one day. (**K**) Abundant barnacle secretions are found among glass beads [55]. Copyright 2017, Elsevier.

**Figure 2 ijms-25-07994-f002:**
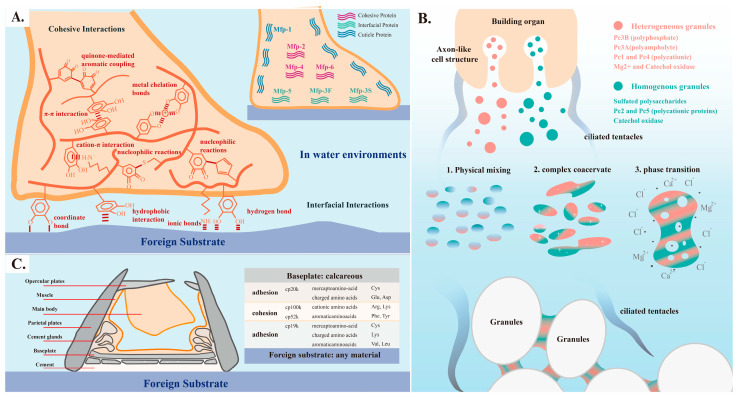
Key aspects of adhesives in mussel plaques, sandcastle worm glue, and barnacle cement: (**A**) Cohesion and adhesion principles of mussel byssal proteins. (**B**) Underwater adhesion process and mechanism of sandcastle worms [7]. (**C**) Distribution and amino acid preferences of barnacle cement proteins.

### 2.4. Summary of Adhesion Processes and Mechanisms

Mussels, barnacles, and sandcastle worms anchor themselves to surfaces exposed to intense seawater currents by secreting bioadhesives with exceptional adhesion abilities. The bioadhesives secreted by these three marine organisms exhibit a high environmental sensitivity. Secreted by specialized cells or glands, they typically maintain a fluidic state under physiological conditions. The liquid bioadhesives rapidly react and solidify upon contact with the marine environment, and they exhibit potent adhesion effects.

Mussel byssal threads primarily adhere to hard surfaces such as rocks and ship hulls. They rely on the plaque-thread system affixed to the interface. The thread is primarily composed of fibers along the axial direction, while the plaque portion exhibits a foam-like porous structure. Sandcastle glue binds sand grains and small particles, and its main structure also displays a foam-like porous morphology. The structure imparts high mechanical strength to the bioadhesives and impedes crack propagation. Barnacle cement adheres extensively to diverse substrates, ranging from hard surfaces (e.g., rocks) to soft surfaces (e.g., whale skin), at its base while firmly adhering to its calcium baseplate at the top. Abundant fibrous structures present in barnacle cement enhance its mechanical strength and resilience at the base.

The cured bioadhesives of these three marine organisms exhibit exceptional stability and minimal swelling even in harsh marine environments. In addition, these three bioadhesives possess a certain level of toughness as they can absorb and disperse external impact energy through deformation, which reduces the occurrence of fractures and damages. Mussel byssal threads exhibit the greatest ductility.

## 3. Underwater Adhesion Processes and Potential Mechanisms in Exemplary Biological Models

The primary components of mussel byssal threads, sandcastle glue, and barnacle cement are proteins. Physical and chemical reactions occurring within bioadhesives, either internally or at interfaces, largely determine their structure and adhesion strength. The investigation into the adhesion mechanisms of diverse marine organisms has yielded various strategies for the development of biomimetic approaches. 

### 3.1. Mussel-Inspired Biomimetic Adhesives Based on DOPA

The formation of mussels’ plaque-thread system involves the participation of three main mussel adhesive proteins (MAPs), which contribute to robust adhesion. Foot proteins (Mfps) exhibit approximately 20 identified variants. Mfp-3F, Mfp-3S, and Mfp-5 are responsible for interfacial adhesion, while Mfp-1 mainly forms the cuticle layer surrounding the interface. The main role of Mfp-2 and Mfp-4 is to promote cohesion within the byssal thread. Therefore, Mfps are crucial for interfacial adhesion. Collagen proteins, including preCol-P, preCol-D, and preCol-NG, are the main components of the core of the byssal thread [59]. The cooperation between byssal thread matrix proteins (PTMP and DTMP) and preCols regulates the elasticity and strength of the byssal thread.

The majority of byssal proteins contain 3,4-dihydroxyphenylalanine (DOPA) residues [59,60,61], which are post-translationally modified from tyrosine. DOPA penetrates the interfacial water film and interacts with the substrate surface, which facilitates strong crosslinking within adhesives. Therefore, it is crucial in the adhesion process [62,63].

Catechol groups (or ortho-dihydroxyphenyl) in DOPA can form covalent bonds through various mechanisms. (1) DOPA–metal coordination: 2–3 DOPA molecules coordinate with a metal ion center to form a bidentate complex with the metal ion center serving as a crosslinking point. DOPA–metal coordination bonds exhibit dynamism and remarkably high stability constants. These properties contribute to the protein’s viscoelastic dissipation and enhanced toughness [64]. Harrington et al. validated that DOPA in the mussel cuticle can form clustered catechol–iron chelates with inorganic ions (especially Fe^3+^) using Raman spectroscopy. This results in a particle-like cuticle structure exhibiting both high hardness (H ≈ 100 MPa) and high ductility (ε_ult_ ≈ 70–100%) [38]. (2) Catechol oxidation: Catechol is a strong reducing agent. The two hydroxyl groups undergo oxidation to form catechol quinone or semiquinone in alkaline conditions (pH > 9) (DOPA-catechol to DOPA-quinone). Quinones readily undergo reactions with electron-rich amino acids through Michael addition or Schiff base reactions due to their polarity and electrophilic sites. They can also undergo phenol radical coupling to form di-DOPA crosslinks. These reactions contribute to the formation of crosslinked network structures or interfacial adhesion [65]. Quinones can also directly generate covalent interactions with the substrate [14]. The oxidation of catechol groups in DOPA is difficult to control despite its enhancement effect on underwater adhesion strength. The strong oxidation tendency disrupts the adhesive layer and diminishes the adhesion strength in alkaline conditions [60,66].

Catechol groups in DOPA facilitate numerous non-covalent interactions, including hydrogen bonding interactions among hydroxyl groups, hydrophobic interactions and π-π interactions provided by hydrophobic benzene rings, and cation-π interactions between amino groups or metal ions and the benzene ring (Figure 2A).

The development of mussel-inspired adhesives primarily relies on the utilization of DOPA. Cheng et al. functionalized GelMA hydrogel with DOPA to obtain an adhesive hydrogel termed GelMA-DOPA. The cohesive strength and adhesion of GelMA are enhanced through the interaction among DOPA molecules. The Lap shear tests of 10% (*w*/*v*) GelMA-DOPA show an adhesion strength of 20.0 ± 1.5 kPa in wet conditions, which is five times higher than that of 10% (*w*/*v*) GelMA [67]. Zhao et al. developed a temperature-responsive bio-inspired adhesive utilizing DOPA as the functional group for adhesion. The pNIPAM side chains collapse above the lower critical solution temperature (LCST), which facilitates the interfacial interaction between DOPA and the target surface. At this point, the underwater adhesion strength is approximately 23 nN (4370 mJ m^−2^), about five times greater than that of 3 M double-sided tape. However, numerous hydrogen bonds are formed between the pNIPAM side chains and water molecules as the temperature decreases. Then, DOPA underneath the swollen pNIPAM side chains is constrained, which leads to adhesive failure [10]. Jenkins et al. polymerized 3,4-dihydroxymandelic acid containing catechol groups with polylactic acid (PLA) to produce a bio-inspired adhesive polymer with a controllable degradation rate. The adhesion strength increases as the proportion of 3,4-dihydroxymandelic acid increases within the range of 0–7%. The phenomenon is attributed to the inter-chain crosslinking and interfacial interaction between the adhesive and the substrate facilitated by catechol groups. The underwater adhesion strength reaches approximately 1 MPa. However, further increased catechol groups cause excessive hydroxyl groups that are bound with water molecules. Consequently, the water absorption of the hydrogel is increased, and its cohesive strength is compromised. The adhesion strength decreases correspondingly. Subsequently, (IO_4_)- is introduced into the polymer gel system, where it oxidizes catechol groups. More covalent interactions are generated, and ultimately the adhesion strength of polymer gels is enhanced by approximately 2–3 times [68].

Incorporating DOPA as the adhesive component into polymer materials is convenient and efficient, yet catechol groups are relatively unstable. Mussels inhibit the oxidation of DOPA by creating an acidic environment and reducing the environment rich in thiol groups in natural environments. Inspired by this principle [69,70], Lee et al. designed environmental changes as a switch to trigger adhesion. They introduced DOPA into a thiol-containing resin material (PETMP) to produce PETMP catechol gels. Thiol groups prevent the oxidation of catechol groups at a low pH. Quinone-based crosslinking and catechol-mediated metal chelation are initiated with increased pH, which enhances adhesion. The combination of one catechol residue with poly (methyl methacrylate) (PMMA) results in the formation of PETMP mono-catechol gels, with an adhesion strength of approximately 0.2 MPa (tensile strength of ~0.2 MPa). Adhesion strength reaches approximately 0.4 MPa when each PETMP binds two catechol residues [64].

### 3.2. Sandcastle Worm-Inspired Biomimetic Adhesives Based on Coacervation and Phase Transition

Sandcastle glue is characterized by its high protein and metal salt content, as well as the presence of sulfated polysaccharides and enzyme tyrosinase. Six adhesive proteins have been identified in sandcastle glue. Pc1, Pc2, Pc4, and Pc5 are polybasic, while Pc3A is polyacidic. Pc3B is a zwitterionic polyelectrolyte. Tyrosine residues in Pc1 and Pc2 are post-translationally modified to DOPA, while substantial serine residues in Pc3A are phosphorylated to phosphoserine [7]. 

The complex coacervation and two-step curing processes are the primary mechanisms responsible for the underwater adhesion of sandcastle glue [7,71]. The physicochemical process arises from the mixing of homogeneous and heterogeneous secretion granules. The polybasic cationic proteins Pc2 and Pc5, together with sulfated polyanions, achieve charge balance within homogeneous secretion granules. The polybasic cationic proteins Pc1 and Pc4, polyphosphate protein Pc3B, and polyampholyte protein Pc3A achieve charge balance within heterogeneous secretion granules [45]. Proteins with opposite charges undergo complexation driven by electrostatic interactions when the homogeneous and heterogeneous secretion granules are physically mixed near micropores. Then, complex coacervate phases are formed. These newly formed coacervate phases are particularly sensitive to ion strength and pH values. The phases rapidly bind with Mg^2+^ and Ca^2+^ upon transition from a physiological environment (pH < 6) to a seawater environment (pH > 8). The initial curing process is completed within approximately 30 seconds. Subsequently, the oxidation of DOPA in Pc1 and Pc2 by tyrosinase leads to the crosslinking and aggregation of coacervates, indicating the completion of the second curing process (Figure 2B) [46,72].

Complex coacervates in natural environments undergo curing as a result of changes in environmental conditions, which involve extensive covalent and non-covalent interactions. Inspired by this, researchers developed polyelectrolyte material systems based on coacervation and phase transitions. These systems can trigger crosslinking reactions and enhance toughness in response to external environmental changes, such as alterations in the pH, ionic strength, temperature, and solvent [73,74,75].

Zhao et al. prepared a solvent-exchange-triggered wet-adhesive by dissolving catechol-modified polyacrylic acid (PAAcat) and quaternized chitosan (QCS-Tf2N) in dimethyl sulfoxide (DMSO). Solvent exchange occurs upon the injection of the polymer blend solution into water as a result of the miscibility difference between DMSO and water. The phenomenon results in the formation of aggregates on glass surfaces within 2 min, which is facilitated by electrostatic interactions between the negatively charged PAAcat and positively charged QCS-Tf2N. The adhesion strength rapidly rises to approximately 200 mN within 10 min. The aggregates are resistant to displacement by hydrodynamic shear forces and can withstand water jet impact up to 30 bar for 1 h due to their highly hydrated surfaces. Ultimately, an adhesive work (W_adh_) of approximately 2 Jm^−2^ is achieved [74].

Marco et al. prepared a thermos-responsive composite coacervate by grafting poly (N-isopropylacrylamide) (PNIPAM) onto a polyelectrolyte scaffold with opposite charges. PNIPAM remains expanded below the LCST, which hinders the interaction between the negatively charged PAA-g-PNIPAM and positively charged PDMAPAA-g-PNIPAM. At this point, the material exhibits a liquid-like state with a W_adh_ of approximately 0.02 Jm^−2^. PNIPAM contracts and aggregates when the temperature exceeds the LCST, which facilitates electrostatic interactions between positively and negatively charged segments. Physical crosslinking reaches a W_adh_ of approximately 1.6 Jm^−2^. The thermos-responsive composite coacervate demonstrates a higher W_adh_ compared to standard commercial pressure-sensitive adhesives. Additionally, reversible crosslinks enable the reconstruction of physical bonds through a cooling process after adhesive failure [73].

### 3.3. Barnacle-Inspired Biomimetic Adhesives Based on Proteins’ Multiple Interactions and Self-Assembling

Barnacle cement proteins (CPs), the main components of barnacle cement with well-defined hierarchical structures, are categorized into seven types based on their molecular weight [26,76,77]. CP20k and CP19k are interfacial proteins. CP19k, predominantly located at the base of barnacle cement, is accountable for the adhesion between barnacle cement and the interface. Studies on recombinant CP19k proteins reveal their preferences for amino acids that are charged, hydrophilic, and hydrophobic. Consequently, the strong adhesion of CP19k to charged or hydrophobic materials is attributed to the synergistic effect between cationic lysine and adjacent hydrophobic amino acids, which facilitates electrostatic interactions with negatively charged rock surfaces [78]. Additionally, disordered residues endow CP19k with significant spatial flexibility, which facilitates its adaptation to various substrates. CP20k, primarily located at the top layer of barnacle cement, is responsible for the interfacial adhesion between the main body of barnacle cement and the calcium substrate. It is rich in cysteine, charged amino acids, and anionic residues. The electrostatic repulsion effect endows CP20k with flexibility in spatial conformation, which exhibits strong coordination interactions [79,80]. CP20k and CP19k undergo self-assembly processes that lead to changes in molecular conformation and the generation of nanofiber-like structures when the physiological environment transitions to a seawater environment [78,81].

The proteins CP52k and CP100k, characterized by relatively large molecular weights and abundant fibrous structures, are primarily distributed within barnacle cement to facilitate cohesion. These proteins are enriched with cationic amino acids (arginine (Arg) and lysine (Lys)), aromatic amino acids (phenylalanine (Phe) and tyrosine (Tyr)), and substantial hydrophobic aliphatic residues. The composition promotes the establishment of robust hydrophobic interactions and cation-π interactions, which augment cohesion strength. However, the synergistic mechanism by which interface proteins and cohesion proteins collaborate to achieve robust underwater adhesion is still under investigation (Figure 2C) [49,66,82].

Various functional groups in barnacle cement contribute to the formation of dense and intricate multiple interactions, which enable it to react with diverse substrates. Meanwhile, the maintenance of stability against oxidation, enzyme degradation, and other challenges encountered in complex environments is ensured. The combination of these facts leads to an enhanced adhesion efficiency. Additionally, the abundant self-assembled fibrous structures enhance the mechanical strength of barnacle cement and consolidate the layered structure into a cohesive entity through crosslinking among fibers [57]. Extensive biomimetic adhesives are designed based on the characteristics of multiple interactions and self-assembled nanofibrous structures in barnacle cement.

Inspired by the abundant cation-π interactions, FAN et al. utilized ATAC (2-(acryloyloxy) ethyl trimethylammonium chloride) with cationic properties to mimic the numerous cationic amino acids in barnacle cement. They also employed PEA (2-phenoxyethyl acrylate), a hydrophobic compound containing a benzene ring, to simulate the hydrophobic amino acids prevalent in barnacle cement. A bioinspired hydrogel is synthesized by combining these components. The hydrophobic PEA disrupts the hydration layer on the substrate surface, which allows for the formation of hydrophobic bonds at the interface. Meanwhile, the positively charged ATAC establishes electrostatic interactions with negatively charged solid surfaces underwater. Furthermore, the benzene ring in PEA and the ammonium ions (NH^4+^) in ATAC facilitate π-π interactions and cation-π interactions, respectively, which further enhance interfacial adhesion and the hydrogel’s cohesion. The barnacle-inspired hydrogel exhibits an adhesion strength of up to 180 kPa, an elastic modulus of 0.35 MPa, a fracture stress of 1.0 MPa, and a fracture strain of 720%. In addition, these properties can be maintained for several months underwater [66].

Liang et al. expressed the recombinant protein rBalcp19k in Escherichia coli based on the nano-fibrous microstructure of proteins in barnacle cement. The secondary structure of rBalcp19k is mainly composed of random coils and β-sheets. rBalcp19k can self-assemble into thioflavin T-insensitive nano-fibers in conditions of acidity and low ionic strength. The self-assembled rBalcp19k exhibits an adhesion strength comparable to that of commercial Cell-Tak adhesives using a colloidal probe technique based on atomic force microscopy (AFM) [57]. However, researchers have shifted their focus toward peptide investigations due to the considerable challenges posed by studying full-length proteins [82,83].

### 3.4. Development of Other Biomimetic Adhesives and Applications

The progress of fundamental research on biological models has led to remarkable advancements in mimicking underwater adhesive organisms, which promotes the application of bioinspired adhesives. The preparation process of traditional synthetic adhesives in the biomedical field typically involves the use of toxic crosslinking agents and solvents. These substances can induce toxic reactions in the body or generate pro-inflammatory by-products during degradation, which results in local inflammation and tissue damage [84,85]. In contrast, bioinspired adhesives mimic the adhesion mechanisms of natural organisms. They are typically prepared under mild conditions to avoid the use of toxic crosslinking agents. Additionally, the base materials of bioinspired adhesives are derived from natural molecules, which are biodegradable in the body and can produce non-toxic degradation products. These characteristics reduce tissue irritation and immune rejection [86,87,88]. Therefore, bioinspired adhesives exhibit significant advantages in biomedical fields due to their excellent biocompatibility.

Oral drug delivery is clinically needed [89]. Oral dosage forms must maintain prolonged contact with the oral mucosa to ensure effective drug delivery. The maintenance of strong adhesion is crucial for achieving effective drug delivery in the moist environment of the oral cavity. Hu et al. prepared a thin-film buccal tissue adhesive using poly (vinyl alcohol) (PVA) and DOPA. The catechol groups in the adhesive can form hydrogen bonds with the PVA matrix, as well as physical and covalent bonds with oral mucus. Subsequently, drug-loaded PLGA-PDA nanoparticles are surface-bound on the film. The adhesion properties and mechanical strength of the modified film can be optimized to match those of commercial films by adjusting the ratio of DOPA. Additionally, the adhesion of DOPA to nanoparticles and film enables a slow, controllable, and sustained release of the drug from the nanoparticles. Then, the released drug is taken up by epithelial cells. The buccal drug delivery using the mucoadhesive film and drug-loaded nanoparticles exhibits an enhanced efficiency of drug absorption compared to oral administration, according to pharmacokinetic studies conducted in Sprague Dawley rats [90].

The utilization of liquid emboli in clinical use is accompanied by significant limitations, including the risk of catheter occlusion, chemical injury, and the release of toxic solvents upon phase separation. Inspired by the complex coacervation of sandcastle glue, Jones et al. utilized polycationic salmine sulfate (Sal) and polyanionic phytic acid (IP) to prepare oppositely charged polyelectrolyte (PE) composite coacervates, termed Sal-IP6. The substance serves as an embolic agent to block blood flow. The Sal-IP6 embolic agent appears as a clear, homogeneous liquid when injected into the rabbit’s kidney through a catheter containing a 1.2 M sodium chloride solution. A sudden change in ion concentration causes rapid coacervation of the polyelectrolyte upon removal of the catheter. Imaging reveals precise and complete uniform embolization of the rabbit’s left renal artery, devoid of alterations in respiration, heart rate, or pain during or after the injection. This ion concentration-controlled embolic agent can partially mitigate the risks associated with traditional embolization techniques, such as chemical injury, catheter occlusion, and biological toxicity [91].

Kim et al. utilized blood or water as a trigger to initiate the complex coacervation involving anionic hyaluronic acid (HA) and a recombinant mussel adhesive protein (rMAP) that contains tyrosine residues. The process is inspired by the adhesion of DOPA in mussels and the complex coacervation of sandcastle worms. Xenogeneic bone’s substitute materials, deproteinized bovine bone minerals (DBBM), are stably encapsulated within the coacervate. The resulting hydrogel immobilizes DBBM at the defect site, which prevents ectopic bone formation. Additionally, it promotes platelet adhesion for hemostasis purposes and combats bone loss caused by blood or bodily fluids [92]. 

The time-consuming nature of suturing incisions constitutes a drawback, while local hemostatic agents frequently exhibit inadequate hemostasis during wound healing. Traditional hemostatic adhesives typically require consistent pressure application or UV irradiation, both of which present limitations in clinical use. Researchers, drawing inspiration from the bioadhesives secreted by underwater adhesive organisms, developed biomimetic tissue adhesives that demonstrate excellent adhesion properties for hemostasis purposes [93]. Yuk et al. simulated the biological mechanism by which barnacles secrete a lipid-rich matrix to clean the substrate and then form barnacle glue proteins for attachment. They used silicone oil to simulate the lipid-rich matrix and employed polyacrylic acid-N-hydroxysuccinimide ester (PAA-NHS ester) to mimic the adhesive component of barnacle glue proteins. Then, the two substances are mixed to form a composite paste. The paste is applied by compression upon usage. The lipid-rich matrix cleans the tissue surface covered with blood within seconds and forms transient hydrogen bond interactions. The ester groups of NHS form more stable covalent bonds with the abundant amine groups in the biological tissue after several minutes. The biomimetic adhesives exhibit exceptional adhesion properties, with shear and tensile strengths of 70, 55, 45, and 50 kPa when adhered to skin, aorta, heart, and muscle tissue, respectively. Furthermore, they show significant improvements in cell activity, inflammation level, and rejection reaction compared to commercial adhesives like CoSeal. These improvements make them an ideal choice for dynamic tissues [51].

Introducing underwater adhesive groups onto the interface of commercial flexible electrode materials facilitates their functionality in aquatic environments. In addition, integrating these electrodes with wearable devices allows for the underwater monitoring of biological signals originating from the skin [94,95]. Ji et al. developed an underwater biomimetic polymer, that is, p(DMA-co-AA-co-MEA) (pDAM). It utilizes dopamine methacrylamide (DMA) containing catechol groups for adhesion, acrylic acid (AA) for conductivity in water, and 2-methoxyethyl acrylate (MEA) for the formation of a stable scaffold underwater. The combination of pDAM and a stretchable electrode of Au/PDMS establishes a pDAM/Au/PDMS electrode, which achieves stable adhesion to the skin surface and signal transmission simultaneously. The electrocardiogram (ECG) signals from the pDMA/Au/PDMS electrode remain stable throughout a 60 min immersion period in water based on experimental results [94].

Biomimetic adhesives are applicable in biomedical and engineering sectors, such as underwater monitoring [96], underwater adhesion [21,97], membrane-based water treatment [98], and anti-fouling [99,100]. In these engineering domains, adhesives that can function in aquatic environments are paramount. 

Table 1 lists the applications of biomimetic adhesives in aquatic environments presented in references. In particular, research on biomimetic adhesives based on mussels’ DOPA is relatively abundant. Biomimetic adhesives in aquatic environments benefit from their extensive non-covalent interactions, which exhibit remarkable reversible adhesion properties. Additionally, their excellent biocompatibility and relatively controllable degradation properties lead to significant progress in fields such as drug delivery, tissue adhesion, and tissue engineering. Furthermore, their superior adhesion properties and resistance to swelling in aquatic environments provide innovative strategies for addressing engineering adhesion dilemmas. Therefore, biomimetic adhesives have been extensively explored in fields such as smart wearable devices, underwater monitoring, and membrane-based water treatment. Predictably, these adhesives designed for aquatic environments hold significant application potential in life signal transmission, environmental monitoring, and underwater robotics. Their in-depth research is worthwhile.

Extensive biomimetic materials have been designed based on the adhesion properties of three marine organisms. High-molecular-weight polymers with carbon chains as the backbone are commonly used in biomimetic designs inspired by these organisms.

The adhesion mechanism of mussels is most extensively studied, which promotes the development of high-performance biomimetic adhesives based on DOPA groups. However, DOPA groups are susceptible to oxidation, which leads to unpredictable alterations in the adhesive effect. Researchers designed some biomimetic adhesives with oppositely charged polyelectrolytes based on the complex coacervation and two-step curing mechanism of sandcastle glue. These adhesives primarily initiate adhesion or undergo controllable repair of the coacervate-to-solid transition by changing the conditions of the aquatic environment. However, the high osmotic pressure within the adhesives can result in significant swelling, which reduces their mechanical strength. The comprehensive understanding of the underlying mechanism of barnacle cement remains elusive despite its superior adhesion properties in aquatic environments. The robust cohesion and interfacial adhesion effect of barnacle cement are related to the interactions among various amino acids and the formation of nanofibrous structures based on current research. Similarly, barnacle-inspired adhesive designs primarily involve the utilization of high-molecular-weight polymers featuring multiple interactions and nanofibrous structures.

## 4. Conclusions and Perspectives

Traditional adhesives encounter challenges in aquatic environments, such as reduced adhesion, susceptibility to swelling, and potential leaching toxicity. In contrast, bioadhesives secreted by marine organisms (e.g., mussels, sandcastle worms, and barnacles) exhibit strong adhesion underwater and maintain stability in harsh marine conditions with minimal volume changes. 

Mussel byssal threads, sandcastle glue, and barnacle cement exist in a liquid form in physiological environments during the initial secretory stage. They are rapidly activated and solidified upon contact with seawater, which results in stable adhesion. The plaque is primarily responsible for the adhesion in mussel byssal threads. Mussel byssal threads and sandcastle glue exhibit similar foam-like porous structures. Barnacle cement uniquely contains abundant fibrous structures. The two distinct structural patterns are crucial for their respective mechanical strength and adhesion abilities. Additionally, mussel byssal threads possess a fibrous structure within the core, which contributes to their ductility.

A series of bioadhesives has been designed based on the adhesion mechanisms of mussel byssal threads, sandcastle worm glue, and barnacle cement. 

Current research on mussels has acknowledged the pivotal role of catechol groups in DOPA during adhesion. Therefore, extensive research has been conducted on biomimetic adhesion using DOPA. However, the oxidation and covalent crosslinking of DOPA are more prone to occur in alkaline environments or with oxidants. The phenomenon poses challenges in the control of adhesion processes. Therefore, the key to the development of mussel-inspired biomimetic adhesives lies in controlling the reaction direction of the functional groups in DOPA. Consequently, stable biomimetic adhesives can be obtained.

The adhesion of sandcastle glue primarily relies on the coacervation and phase transition processes of oppositely charged electrolytes. Inspired by the mechanism, biomimetic adhesive designs mainly focus on the formation of adhesive hydrogels from polyelectrolyte complexes under rapidly changing environmental conditions. The adhesion performance of synthetic coacervates is influenced by environmental pH or ionic strength due to their charged units. Therefore, adhesives for specific environments can be customized.

The exceptional adhesion efficiency of barnacle cement is attributed to the complex interactions among functional groups in proteins that can self-assemble into nanofibers. However, the complex composition and structure of barnacle cement proteins pose challenges in comprehensively investigating their adhesion mechanisms. Practical applications of barnacle cement are limited as a result. Therefore, in-depth theoretical research on nanofiber formation and the synergistic effect of multiple functional groups is essential for the development of biomimetic adhesives with nanofiber structures.

The main components of these three bioadhesives are proteins. The biomimetic designs for these bioadhesives mainly focus on polymer materials with carbon chains serving as the backbone. Looking forward, the utilization of proteins or peptide chains as main components in direct biomimetic designs shows great potential for significantly enhancing the biocompatibility of biomimetic adhesives.

In conclusion, this work summarized the practical applications of biomimetic adhesives and explored potential research directions by considering their excellent adhesion performance. Biomimetic adhesives have gained attention in biomedical fields (e.g., wound healing, dental restoration, and bone tissue engineering) due to their ideal biocompatibility. Moreover, their applications in engineering fields (e.g., wearable devices, underwater adhesion, and membrane-based water treatment) are continuously expanding. The development of biomimetic adhesives that can rival natural bioadhesives requires further explorations on their internal cohesion, interfacial adhesion, and responsiveness to stimuli.

## Figures and Tables

**Table 1 ijms-25-07994-t001:** Biomimetic adhesives in aquatic environments.

	Materials	Biological Model	Application Fields	Effect	Ref.
1	Mixing of N-hydroxysuccinimide modified poly (lactic-co-glycolic acid) nanoparticles (PLGA-NHS) and alginate-dopamine polymer (Alg-Dopa)	Mussel	Biodegradable tissue adhesive	Lap shear strength of 33 ± 3 kPa for porcine skin-muscle interface; degradable; cytocompatible; minimal inflammatory responses.	[101]
2	Poly (ethylene glycol) diacrylate/alginate double network hydrogels and 3,4-dihydroxy-L-phenylalanine as a crosslinker	Mussel	Skin dressings	High mechanical strength and self-healing properties with a highly transparent appearance.	[102]
3	Poly (acrylamide-co-dopamine) with lithium chloride	Mussel	Flexible strain sensors	Self-healing, stretchable, adhesive, and conductive.	[103]
4	Multipotent flap-protective adhesive mangiferin (MF)-loaded liposomes (A-MF-Lip)	Mussel	Local drug delivery for promoting the generation of skin flaps	Liposomes exhibit great adhesion properties, and adherent MF-loaded liposomes possess multipotent flap-protective therapeutic effects such as pro-neovascularization, cytoprotection, anti-apoptosis, and anti-inflammatory.	[104]
5	Chitosan-graft-L-lysine-L-DOPA	Mussel	Fragrance delivery systems in personal care products	CLD can facilitate the deposition of biodegradable fragrance carriers on diverse surfaces, including hair, cotton, and skin.	[105]
6	Catechol-modified polyacrylamide	Mussel	Reservoir fracture control	Excellent reservoir adaptability (96 °C; 4.7 × 10^4^ mg/L); capable of withstanding water flushing and maintaining stable adhesion to the fracture wall to guarantee the long-term control effect.	[106]
7	Mixing HB-PBAE, poly (1-vinylimidazole) (PVI), and gelatin solution, followed by adding Fe^3+^	Mussel	Wound-healing dressings	Capable of accelerating the wound-healing process and rapidly reducing adhesion; the strength is significantly enhanced upon the spraying of the Zn^2+^ solution.	[107]
8	PVA-DOPA-Cu^2+^ (PDPC) hydrogel	Mussel	Wound healing	Tissue adhesive, antioxidative, photothermal, antibacterial, and hemostatic	[108]
9	Catechol functional groups (DOPA) are crosslinked with the synthetic oligomer oligo [poly (ethylene glycol) fumarate] (OPF)	Mussel	Bone tissue engineering	Capable of enhancing the pre-osteoblast cell attachment and proliferation; DOPA-mediated interfacial adhesive interactions prevent the displacement of scaffolds.	[109]
10	GelMA-PDA hydrogel with TGF-β3 as a cartilage repair layer; GelMA-PDA/HA hydrogel with BMP-2 as a subchondral bone repair layer	Mussel	Bone tissue engineering	The hydrogel exhibits a bone area ratio of 65% in a rabbit’s knee joint with full-thickness cartilage defect.	[110]
11	PUP-PPG-DBHP	Mussel	Underwater engineering field	The adhesive can be applied underwater directly, reaching a bonding strength of approximately 1.2 MPa within around 30 s on glass substrates.	[111]
12	Poly (LAEMA-co-GMA-co-BA)	Mussel	Coating materials	The coated surfaces exhibit flatness, smoothness, great antibacterial adhesion properties, and low cytotoxicity.	[112]
13	Poly (TEG-co-CAG)n	Mussel	Antifouling	Polymer-coated surfaces exhibit reduced protein adsorption and a decreased cell count when compared to the control group.	[113]
14	PAHDP	Mussel	Drug delivery	The PAHDP hydrogel, with excellent adhesion properties and safety profiles, can deliver over 10 types of drugs, especially small-molecule drugs.	[114,115]
15	Dense coacervates formed by aminated collagen and phosphodopa copolymer at 25 °C	Sandcastle worm	Craniofacial reconstruction	The adhesive can maintain 3D bone alignment in freely moving rats over a 12-week indwelling period, and it is degradable.	[116,117]
16	Amine-terminated DbaYKY tripeptide links to functionalized molecules	Sandcastle worm	Synthesis of functional hydrogels	The modified hydrogel possesses biological functions such as cell adhesion, antibacterial, and wound repair.	[118]
17	Phytic acid (PA) as the crosslinker for magnesium oxychloride cement (MOC)	Sandcastle worm	The research of magnesium oxychloride cement (MOC)	The integration of phytic acid improves the water resistance, workability, and applicability of MOC, and it is environmentally friendly.	[119]
18	Oppositely-charged polyelectrolytes (PEI and PAA) and catechol-functionalized cellulose nanofibers (TA-CNF)	Sandcastle worm	Medical adhesion	Capable of absorbing fluids and transforming into a hydrogel (<3 s) with great ductility (~14 times its original form), self-healing ability, and an efficient drug-loading capacity.	[120]
19	Poly (glycerol sebacate)-acrylate nanoparticles	Sandcastle worm	Tissue adhesion	Capable of quickly assembling viscous glue.	[44]
20	PC4/Cultrex hybrid hydrogel	Sandcastle worm	Hydrogels for the formation of liver spheroids	Capable of enhancing HepG2 cells to form spheroids and hepatic differentiation.	[121]
21	3-(acrylamidophenyl) boronic acid (AAPBA) and N-2-hydroxyethyl acrylamide (HEAA)	Sandcastle worm	Responsive reversible wet adhesion	Capable of acquiring pH-responsive reversible adhesion.	[122]
22	Multidentate organophosphate, quaternized cellulose, and perfluorinated sulfonic acid are assembled onto polyethersulfone (PES) substrate	Sandcastle worm	Membrane-based water treatment	The water permeance is 93.3 L m^−2^ h^−1^ bar^−1^ with a rejection rate to organic dyes ranging from 90.0 to 99.9%.	[123]
23	Quaternized chitosan and alginate are mixed with various solid materials (nLCBMs/±)	Sandcastle worm	Building material	Excellent mechanical performance (compressive elastic modulus of nearly 400 MPa), recyclability, anti-weathering property, and scalability.	[124]
24	Tyramine-ammonium polyphosphate (TA-APP) serves as an adhesive along with vinyl ester resin to bond with carbon fibers	Sandcastle worm	Functional material	The material possesses broad-spectrum antibacterial and anti-algae capabilities, in addition to a superior flame-retardant effect.	[125]
25	DOPA-rich ELP	Sandcastle worm and mussel	Biomedical glue	It exhibits adhesion strengths of ∼240 MPa in wet environments and >2 MPa in dry environments and is capable of coacervating in physiological conditions.	[126]
26	Grafting catechol and bis-phosphoric acid groups to the polyoxetane backbone	Sandcastle worm and mussel	Underwater bonding	A bonding strength of 0.35 MPa is achieved under humid conditions.	[127]
27	IMglue-SiO_2_(TiO_2_/SiO_2_)2 SH coating	Sandcastle worm, mussel, and lotus leaf	Tissue closure	Antibiofouling, durable, biocompatible, and antithrombotic.	[128]
28	Reduced sericin-tannic acid (rSer-TA)	Barnacle and mussel	Wound healing in vivo and the sealing of fluid leakage in vivo	A bonding strength of >0.1 MPa for tissues and >0.5 MPa for solid plates.	[129]
29	Aromatic, ionic moieties, and nonpolar functionalized copolymer films	Barnacle and mussel	Potential applications in biomedicine or engineering	The wet contact adhesion is ~15.0 N/cm^2^ in deionized water and ~9.0 N/cm^2^ in seawater at a pH of approximately 7.	[130]
30	A composite composed of a silk fibroin (SF) solution and polydopamine (PDA)	Barnacle and mussel	Underwater adhesion	The synthesis of polymers is simple, characterized by a completely biological composition. A high adhesion strength (>2 MPa) can be achieved using a relatively low mass (1–2 mg).	[131]
31	PEI and PMAA	Barnacle	Hydrogel for adhesive	High mechanical strength (2.66 ± 0.18 MPa) and adhesion strength (1.99 ± 0.11 MPa under water and 2.70 ± 0.21 MPa under silicon oil).	[132]
32	Coating RGD-containing peptides on a polystyrene plate	Barnacle	Tissue engineering scaffolds	Capable of facilitating cell adhesion and spreading.	[133]
33	Poly (LAEMA-co-GMA-co-BA)	Mussel	Coating materials	The coated surfaces exhibit flatness, smoothness, great antibacterial adhesion properties, and low cytotoxicity.	[112]
34	Mrcp19k-inspired low-complexity STGA-rich adhesive peptides (Mr-AP1 and Mr-AP1C)	Barnacle	Underwater adhesion	The adhesive peptides generate adhesive patches under conditions of low pH and low ionic strength.	[134]
35	Prepared MXene/PHMP hydrogel using PEA, MEA, and HEAA in the presence of conductive MXene nanosheets	Barnacle	Underwater sensing	It exhibits rapid and reversible adhesion with minimal swelling, which facilitates the manufacturing of stable and sensitive underwater sensors.	[135]

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
