# Peer review of "An Overview on the Adhesion Mechanisms of Typical Aquatic Organisms and the Applications of Biomimetic Adhesives in Aquatic Environments"

_ijms, 2024, doi:10.3390/ijms25147994_

Round 1

Reviewer 1 Report

Comments and Suggestions for Authors

In the paper, the underwater adhesion processes of mussels, sandcastle worms, and barnacles are summarized, focusing on the structure and components of their secreted biological adhesive. By analyzing the physicochemical processes of adhesion formation in these three marine organisms, the adhesive characteristics of these biological adhesives are summarized. A vast array of biological adhesive designs based on these three marine organisms is then introduced, emphasizing the correlation between adhesive effects and the environment in aquatic settings. The work was well written.

1) More deep comparison could be given in the first part. 

2) The bottom up art could be summarized for the DOPA modified surface, membrane and reaction. 

Author Response

Point-by-point response to Comments and Suggestions for Authors

Comments 1: More deep comparison could be given in the first part. 

Response 1: Thank you for pointing this out. We agree with this comment. Consequently, we have revised the summary of the first section to make the conclusions more comprehensive, specifically in section 2.4 on page five. Below is the updated text:

“Mussels, barnacles, and sandcastle worms anchor themselves to surfaces exposed to intense seawater currents by secreting bioadhesives with robust adhesion capabilities. The bioadhesives secreted by these three marine organisms are highly sensitive to their environment. Secreted by specialized cells or glands, they typically remain in a fluid state under physiological conditions. Upon contact with the marine environment, the liquid bioadhesives rapidly react and solidify, exhibiting potent adhesion effects.

Mussel byssal threads primarily adhere to hard surfaces such as rocks and ship hulls. They rely on the plaque-thread system attached to the interface. The thread is primarily composed of fibers along the axial direction, while the plaque portion exhibits a foam-like porous structure. Sandcastle glues bind sand grains and small particles together, and their main body also displays a similar foam-like porous structure. This structure imparts high mechanical strength to the bioadhesives and can impede crack propagation. Barnacle cement adheres broadly to diverse substrates, ranging from hard surfaces like rocks to soft surfaces like whale skin, at its base while firmly adhering to its own calcium baseplate at the top. Barnacle cement is rich in fibrous structures, which enhance the mechanical strength and resilience of the the bottom of barnacle cement.

Cured bioadhesives of these three marine organisms remain stable even under harsh marine environment, exhibiting minimal swelling. Additionally, these three bioadhesives possess a degree of toughness, as deformation of the adhesives can absorb and disperse external impact energy, reducing fracture and damage. Among them, mussel byssal threads has the greatest ductility.”

Comments 1: The bottom up art could be summarized for the DOPA modified surface, membrane and reaction. 
Response 2: We are not quite clear about this comments. Would you please be more specific? Thank you.

Reviewer 2 Report

Comments and Suggestions for Authors

The review article by Biru Hu and coworkers deals with a hot topic, i.e. biomimetic adhesives, inspired from those secreted by aquatic organisms. Three marine organisms were studied in particular, e.g. mussels, sandcastle worms and barnacles. The structure of the review is linear, clear and valuable, although a bit too long. However, the organization in sections allows the reader to skip some content and focus on the desired information.

It could be useful to highlight, more than the Authors already did, the need to avoid toxic crosslinkers and solvents during the development of biomimetic adhesives, prefering biocompatible molecules and avoiding the release of proinflammatory byproducts from the adhesives.

Check the ending of the conclusion section.

Comments on the Quality of English Language

Please carefully revise English language throughout the manuscript. Minor corrections are needed, even though the paper is understandable in all its sections.

Author Response

Point-by-point response to Comments and Suggestions for Authors

Comments 1: It could be useful to highlight, more than the Authors already did, the need to avoid toxic crosslinkers and solvents during the development of biomimetic adhesives, prefering biocompatible molecules and avoiding the release of proinflammatory byproducts from the adhesives.
Response 1:Thank you for pointing this out. We agree with your comment and have addressed it by adding a section explaining why bio-inspired adhesives outperform synthetic adhesives. This discussion highlights the advantages and underlying reasons for the superior performance of bio-inspired adhesives. The updated text can be found in Section 3.4 on page 10 of the revised manuscript. Below is the updated text:

“As fundamental research on biological models progresses, researchers have made exciting advancements in mimicking underwater adhesive organisms, driving the application of bioinspired adhesives. In the biomedical field, traditional synthetic adhesives often use toxic crosslinking agents and solvents during the preparation processes. These substances can cause toxic reactions in the body or release pro-inflammatory by-products during degradation, leading to local inflammation and tissue damage [86, 87]. In contrast, bioinspired adhesives mimic the adhesion mechanisms of natural organisms and are typically prepared under mild conditions, avoiding the use of toxic crosslinking agents. Additionally, bioinspired adhesives use natural molecules as base materials, which are biodegradable in the body and produce non-toxic degradation products, reducing tissue irritation and immune rejection [88-90]. Due to their excellent biocompatibility, bioinspired adhesives exhibit significant advantages in biomedical fields.”

Comments 2: Check the ending of the conclusion section.

Response 2: We have addressed your suggestion by adding a summary section to the "Conclusions and Perspectives" part of our manuscript. This new section provides a comprehensive overview of the first major part of the paper, specifically summarizing the structure and function of mussel byssal threads, sandcastle glue, and barnacle cement. The updated text can be found on page 16, lines 576-584 of the revised manuscript. Below is the updated text:

“In their secretory initial stages, mussels byssal threads, sandcastle glues, and barnacle cement are all in liquid form under physiological environments. When they go into contact with seawater, they are rapidly activated and solidified, forming stable adhesion. The plaque primarily responsible for adhesion in mussel byssal threads, as well as sandcastle glues, exhibit similar foam-like porous structures, whereas barnacle cement uniquely contains abundant fibrous structures. Both of these distinct structural patterns are crucial for their respective mechanical strength and adhesion capabilities. Additionally, the mussels byssal threads possess a fibrous structure within their core, which contributes to their certain degree of extensibility.”

Comments 3: Please carefully revise English language throughout the manuscript. Minor corrections are needed, even though the paper is understandable in all its sections.

Response 3: Thank you for pointing this out. We agree with this comment and will embark on embellishing the entire manuscript in our next round of revisions.
